# *TERT* Promoter Mutations and the 8th Edition TNM Classification in Predicting the Survival of Thyroid Cancer Patients

**DOI:** 10.3390/cancers13040648

**Published:** 2021-02-05

**Authors:** Jun Park, Sungjoo Lee, Kyunga Kim, Hyunju Park, Chang-Seok Ki, Young Lyun Oh, Jung Hee Shin, Jee Soo Kim, Sun Wook Kim, Jae Hoon Chung, Tae Hyuk Kim

**Affiliations:** 1Division of Endocrinology & Metabolism, Department of Medicine, Thyroid Center, Samsung Medical Center, School of Medicine, Sungkyunkwan University, Seoul 06351, Korea; jun113.park@samsung.com (J.P.); hj1006.park@samsung.com (H.P.); swkimmd@skku.edu (S.W.K.); thyroid@skku.edu (J.H.C.); 2Department of Digital Health, Samsung Advanced Institute for Health Sciences & Technology, School of Medicine, Sungkyunkwan University, Seoul 06351, Korea; ldh1223@g.skku.edu (S.L.); kyunga.j.kim@samsung.com (K.K.); 3Statistics and Data Center, Research Institute for Future Medicine, Samsung Medical Center, Seoul 06355, Korea; 4Green Cross Genome, Yongin 16924, Korea; cski@gccorp.com; 5Department of Pathology, Samsung Medical Center, School of Medicine, Sungkyunkwan University, Seoul 06351, Korea; yl.oh@samsung.com; 6Department of Radiology, Samsung Medical Center, School of Medicine, Sungkyunkwan University, Seoul 06351, Korea; helena35.shin@samsung.com; 7Department of Surgery, Samsung Medical Center, School of Medicine, Sungkyunkwan University, Seoul 06351, Korea; js0507.kim@samsung.com

**Keywords:** differentiated thyroid cancer, *TERT* promoter, *BRAF*, prognosis, mortality

## Abstract

**Simple Summary:**

In a cohort study involving 393 patients with differentiated thyroid cancer, *TERT* promoter mutations were found to act as an independent poor prognostic factor based on the American Joint Committee on Cancer (AJCC) tumor-node-metastasis 8th edition (TNM-8) in differentiated thyroid carcinoma (DTC) patients, regardless of the histological types or stage at diagnosis. Since the current AJCC TNM-8 is insufficient to distinguish the risk of mortality in patients with differentiated thyroid cancer, a proposal for a new survival prediction model that includes the *TERT* promoter mutational state is required.

**Abstract:**

Our research group has previously shown that the presence of *TERT* promoter mutations is an independent prognostic factor, by applying the *TERT* mutation status to the variables of the AJCC 7th edition. This study aimed to determine if *TERT* mutations could be independent predictors of thyroid cancer-specific mortality based on the AJCC TNM 8th edition, with long-term follow-up. This was a retrospective study of 393 patients with pathologically confirmed differentiated thyroid carcinoma (DTC) after thyroidectomy at a tertiary Korean hospital from 1994 to 2004. The thyroid cancer-specific mortality rate was 6.9% (5.2% for papillary and 15.2% for follicular cancers). *TERT* promoter mutations were identified in 10.9% (43/393) of DTC cases (9.8% of papillary and 16.7% of follicular cancer) and were associated with older age (*p* < 0.001), the presence of extrathyroidal invasion (*p* < 0.001), distant metastasis (*p* = 0.001), and advanced stage at diagnosis (*p* < 0.001). The 10-year survival rate in mutant *TERT* was 67.4% for DTC patients (vs. 98% for wild-type; adjusted hazard ratio (HR) of 9.93, (95% CI: 3.67–26.90)) and 75% for patients with papillary cancer (vs. 99%; 18.55 (4.83–71.18)). In addition, *TERT* promoter mutations were related to poor prognosis regardless of histologic type (*p* < 0.001 for both papillary and follicular cancer) or initial stage (*p* < 0.001, *p* = 0.004, and *p* = 0.086 for stages I, II, and III and IV, respectively). *TERT* promoter mutations comprise an independent poor prognostic factor after adjusting for the clinicopathological risk factors of the AJCC TNM 8th edition, histologic type, and each stage at diagnosis, which could increase prognostic predictability for patients with DTC.

## 1. Introduction

Differentiated thyroid carcinoma (DTC), which includes papillary thyroid carcinoma (PTC) and follicular thyroid carcinoma (FTC), accounts for 90% of all thyroid cancers. DTC is an indolent tumor with a favorable prognosis, but some patients present with cancer-related death due to the aggressive progression of DTC [1,2,3]. The American Joint Committee on Cancer/Union for International Cancer Control (AJCC/UICC) staging is a classification system developed to describe the extent of disease and predict mortality, based on a tumor–node–metastasis (TNM) scoring system. It is the most commonly used staging system for thyroid cancer and was recently revised from the TNM 7th edition (TNM-7) to the 8th edition (TNM-8) [4,5]. The major changes in TNM-8 include the advanced age cutoff from 45 to 55 years [6]. In addition, minimal extrathyroidal extension (ETE) has been excluded from the T3 definition. Level VII lymph nodes (LNs) were reclassified as central neck LNs, N1 disease was not staged up to stage III, and distant metastases in older patients were changed to stage IVB for DTC [7]. As a result, nearly 30% of DTC patients were downstaged by TNM-8 [8]. After the staging system changed to TNM-8, the predictability of cancer-specific survival (CSS) has improved for DTC and PTC patients, but it has not improved for FTC patients [9]. In addition, Manzardo et al. recently reported that the risk of structural recurrence of DTC patients downstaged by tumor classification in TNM-8 may be overlooked [8], and the concept of a molecular profile that has recently emerged in the prognosis is still missing [10].

Telomerase reverse transcriptase (*TERT*) plays an important role in cell immortality by maintaining telomere length [11]. Somatic mutations in the *TERT* promoter have been detected in dozens of human cancers, including thyroid cancer [12]. Further, these alterations have been identified more frequently in patients with advanced stages and distant metastasis. Mutations in the *TERT* promoter are also associated with tumor aggressiveness and increased recurrence and mortality [13,14,15,16]. *BRAF* mutations were first identified in molecular marker-based risk stratification because they are commonly found in thyroid cancer [17,18]. Previous studies have shown that *BRAF* mutations are linked to increased recurrence and mortality, LN metastasis, ETE, and advanced stage in patients with PTC [19,20]. However, since *BRAF* mutations are found in more than 80% of newly diagnosed PTC cases in Korea and are not usually detected in FTC [21], it is difficult to use these alterations for prognostic prediction in DTC.

A previous study has shown that the inclusion of *TERT* mutation analysis strengthened the prognostic predictability of CSS based on the TNM-7 system for patients with DTC [15]. In the present study, we examined whether the analysis of *TERT* mutations could improve the prediction of mortality for DTC patients based on the TNM-8 system.

## 2. Results

### 2.1. Baseline Characteristics

Of the total 393 DTC patients, 329 (83.7%) were women and 64 (16.3%) were men. The median age at diagnosis was 43 years (range of 16–81 years), and 319 patients (81.2%) were under 55 years of age. The total DTC patient group included 327 (83.2%) PTC patients and 66 (16.8%) FTC patients. The sex ratio, average age, and proportion of patients under 55 were similar between PTC and DTC patients. A total of 364 (310 PTC and 54 FTC) DTC patients received post-operative radioactive iodine (RAI) ablation with a median frequency of 2 (range of 0–11) and median total dose of 130 mCi (range of 0–1400 mCi). The median follow-up duration was 16 years (interquartile range of 14–19 years). The cumulative thyroid cancer-related mortality rates were 5.2% (17/327) for PTC patients, 15.2% (10/66) for FTC patients, and 6.9% (27/393) for all DTC patients. The clinical and genetic characteristics of the DTC and PTC patients are summarized in Table 1.

### 2.2. Associations between TERT Promoter Mutation Status and Clinicopathological Variables

*TERT* mutations were detected in 10.9% (43/393) of DTC patients, 9.8% (32/327) of PTC patients, and 16.7% (11/66) of FTC patients (Table 1 and Table 2). Among the DTC patients, there were significantly more *TERT* mutations in patients ≥ 55-years-old (odds ratio (OR) of 12.34; *p* < 0.001), tumors with ETE (OR of 4.98; *p* < 0.001), cases with distant metastasis (OR of 5.10; *p* = 0.001) and patients with advanced stage at diagnosis (OR of 10.73 for Stage II, 32.60 for Stages III and IV; *p* < 0.001; Table 2). The results were similar in PTC patients, except for the association between *TERT* mutations and tumors with distant metastasis, which was not statistically significant.

### 2.3. Associations between TERT Promoter Mutation Status or Clinicopathological Variables and Thyroid Cancer-Specific Survival

We next conducted analyses of the 10-year survival rate and factors affecting survival in DTC and PTC patients. In the univariate analysis of DTC patients, age (*p* < 0.001), *TERT* mutation status (*p* < 0.001), histological type (*p* = 0.004), ETE (*p* < 0.001), distant metastasis (*p* < 0.001), stage at diagnosis (*p* < 0.001), and tumor size (*p* = 0.014) were significant predictors. In the multivariate analysis of the extended model, age (*p* < 0.001), *TERT* mutation status (*p* = 0.002), and histological type (*p* = 0.028) were significant; in the multivariate analysis of the restricted model, the *TERT* mutation status (*p* < 0.001), histological type (*p* = 0.001), and stage at diagnosis (*p* = 0.003) were independent factors that affected thyroid CSS (Table 3). The 10-year survival rate of DTC patients with wild-type *TERT* was 98.0%, and that of patients with a *TERT* mutation was 67.4%, indicating poor prognosis for patients with *TERT* mutations (Table 3 and Figure 1A). The adjusted HRs (95% CI) of *TERT* mutations were 5.18 (1.81–14.82) in the extended model and 9.93 (3.67–26.90) in the restricted model (Table 3). In addition, in the subgroup analysis conducted based on each stage, the survival rate of patients with *TERT* mutations was significantly lower than that for individuals with wild-type *TERT* (*p* < 0.001 in stage I, *p* = 0.004 in stage II, *p* = 0.086 in stages III and IV; Figure 2). Since subjects in stage II showed a significantly lower survival as compared to stage I (Table 3), we stratified the stage II subjects in those who were downstaged vs. those not downstaged by the TNM 8th edition (Appendix A). The mutant *TERT* had a tendency to be more frequent in downstaged vs. not downstaged (*p* = 0.074, statistically not significant), suggesting that in several of these cases, the downstaging may have led toward an underestimation of cancer aggressivity. Additionally, the frequency of mutant *TERT* was not associated with the cause of downstaging (age vs. tumor classification) among downstaged stage II patients (*p* = 0.682) (Appendix A).

The results in PTC patients were similar to those in DTC patients, but the difference was that sex was a significant factor in the univariate analysis (*p* = 0.026) but not in the multivariate analysis. Moreover, the stage at diagnosis was not significant in the multivariate analysis (Table 4). The 10-year survival rate of patients with mutant *TERT* was 75.0%, which was significantly lower than that for patients with wild-type *TERT* (99.0%). Adjusted HRs (95% CI) of mutant *TERT* were 10.68 (2.36–48.27) in the extended model and 18.55 (4.83–71.18) in the restricted model (Table 4 and Figure 1B). In the sensitivity analysis conducted by applying a backward elimination approach, the HRs (95% CI) of the *TERT* mutation in DTC patients and PTC patients were 4.27 (1.28–14.21, *p* = 0.018) and 14.17 (3.00–67.00, *p* = 0.001), respectively. Thus, this also demonstrated that *TERT* mutations were independent factors related to poor prognosis.

## 3. Discussion

Although TNM-7 has been revised to TNM-8 and prognostic predictability has increased [22,23], there is still no concept of molecular marker-based risk stratification, which has emerged based on prognostic genetic markers such as *BRAF* and *TERT* mutations [23]. In addition, TNM-8 does not distinguish patients who are rapidly deteriorating due to aggressive progression during follow-up among patients with the same initial stage. If DTC patients are pre-classified at the time of diagnosis and selectively treated, we can optimize the distribution of medical resources and pursue the best clinical outcomes. Therefore, efforts to further optimize hazard discrimination have continued, such as the search for new prognostic factors that can be considered in addition to the conventional staging system [24,25].

*TERT* promoter mutations are located at two hotspots on chromosome 5, C228T and C250T. C228T is far more prevalent [26,27], and these two mutations generate a new binding site for the MAPK-dependent E-twenty six transcription factors, which is associated with increased *TERT* expression and telomerase activation [28]. *TERT* mutations were first found in melanoma [26,27] and later reported in thyroid cancers [13]. Many studies have shown that they are associated with tumor aggressiveness and are independent prognostic markers for CSS in DTC [13,14,15,29]. Further studies reported the synergistic effects of coexisting *BRAF* V600E and *TERT* mutations on poor clinical outcomes [30,31,32]. A recent study reported a novel molecular mechanism involving the *BRAF/MAPK/FOS/GABP/TERT* pathway, which involves these two mutations and results in increased *TERT* expression [33]. It is known that *TERT* mutations are more common in tumors with *BRAF* mutations [13,29], which might explain why only two patients had *TERT* mutations without *BRAF* mutations in this study. Therefore, we could not directly compare the survival rate of the group with only *TERT* mutations with that of the group having coexisting *TERT* and *BRAF* mutations. However, we confirmed that the mortality risk increased significantly when a *TERT* mutation was combined with a *BRAF* mutation (Appendix A). Moreover, we analyzed the HRs of *TERT* mutations in various models, such as restricted, extended, and backward elimination-based, and proved the independent effect of *TERT* mutations in all models.

In our previous report, we analyzed the survival rate, including *TERT* mutation status based on TNM-7 [15]. In this study, we performed survival analysis by restaging the 393 DTC patients in the cohort from the previous study using TNM-8 guidelines and by extending the follow-up period to determine whether *TERT* mutations are still a factor associated with poor prognosis. *TERT* mutations in DTC patients were still independent poor prognostic factors with a median follow-up of 16 years, and these results were also found based on subgroup analysis of each TNM-8 stage (Figure 2) and each histologic type (Figure 1 and Appendix A).

Differences from the results of the previous study on DTC were that sex was included as a covariate, but there was no significance, and HRs (95% CI) of histology were reduced in the extension model from 9.27 (2.06–41.72) in TNM-7 to 3.02 (1.13–8.09) in TNM-8. In addition, ETE was not significant (*p* = 0.227), based on multivariate analysis. In PTC patients, age was not significant (*p* = 0.059) but most of the other results were similar to the previous findings.

In addition, we added the RAI total dose as a variable in the multivariate analysis of DTC and PTC patients (Appendix A). Considering RAI treatment, the effect size (HRs and *p*-value) of distant metastasis and stage at diagnosis on CSS decreased, which showed that RAI had a therapeutic effect. The effect size of *TERT* mutation was not affected by the RAI total dose, which is consistent with the results of a recent study showing that *TERT*-mutant thyroid cancer is related to poor RAI therapy responses [34].

There are several limitations to this study. This was a retrospective study and included many previous patients diagnosed before ultrasonographic screening became popular to ensure a sufficient long-term follow-up period. Thus, unlike recent trends, there were fewer patients with small-sized thyroid cancer [35,36]. There was also a concern that the prevalence of *TERT* mutations might have been exaggerated because of the high proportion of patients with large-sized thyroid cancer, but the prevalence of *TERT* mutations was 9.8% in PTC patients and 16.7% in FTC patients, which was similar to that reported in two previous studies (approximately 11% and 17%, respectively) [37,38]. Moreover, due to the small sample size, there was a lack of subgroup analysis for FTC patients.

## 4. Materials and Methods

### 4.1. Patients and Clinicopathological Data

A total of 393 patients who were pathologically diagnosed with DTC, including 327 PTC and 66 FTC cases (including Hurthle cell carcinoma) after thyroidectomy and neck dissection between October 1994 and December 2004, were included in this study; all of them were enrolled in a previous study [15]. All patients received thyrotropin suppression therapy, and 364 patients received post-operative radioactive iodine (RAI) ablation according to standard guidelines [39,40]. A pre-existing cohort in a previous study [15] was restaged based on TNM-8 guidelines through pathology reports and surgical record reviews. Since TNM-8 does not include microscopic ETE, this study counted only gross ETE (strap muscles, subcutaneous soft tissue, larynx, trachea, esophagus, recurrent laryngeal nerve, and prevertebral fascia) as ETE, greatly reducing the number of tumors with ETE compared to that in the previous study that used TNM-7. As the number of patients with stages III and IV was very small after restaging, these patients were combined when performing analysis according to stage.

We conducted mutation analyses by taking one sample from the thyroid cancer tissue of each patient at the Department of Pathology of the Samsung Medical Center. Since this was performed after surgery and RAI treatment, the results did not affect the decision-making process of the physicians. Thyroid cancer-related mortality data were obtained from the Korea National Statistical Office and hospital medical records.

### 4.2. DNA Isolation from Thyroid Cancer Samples

Genomic DNA was extracted from formalin-fixed paraffin-embedded (FFPE) tissue using a Qiagen DNA FFPE Tissue Kit (Qiagen, Germany) according to the manufacturer’s instructions. We prepared 4-μm-thick unstained slides from FFPE tissue, and the pathologists decided that slides with a minimum 75% tumor component could be used for DNA extraction.

### 4.3. Detection of TERT Promoter and BRAF T1799A Mutations

We performed a semi-nested polymerase chain reaction to identify *TERT* promoter mutations, mutant enrichment with 3′-modified oligonucleotide-PCR, and direct sequencing for the detection of *BRAF* T1799A mutations, as described in a previous report [15].

### 4.4. Statistical Analysis

Univariate logistic regression analyses were performed to evaluate the association between *TERT* mutation status and clinicopathological variables in patients with DTC. In the survival analysis, the follow-up duration was defined as the time from initial surgical treatment to the date of thyroid cancer-specific death for deceased individuals or the date of last observation (31 December 2018) for survivors. The factors related to thyroid CSS were analyzed by multivariate Cox regression, and only predictors with a *p*-value ≤ 0.2 in univariate analysis were included in the multivariate analysis. The results of univariate analysis were displayed by a Kaplan–Meier survival curve, and hazard ratios and *p*-values were calculated from Cox proportional models. Since the “stage at diagnosis” is a composite variable, we analyzed the multivariate Cox regression by dividing it into two models. In the restricted model, “stage at diagnosis” was analyzed as one factor. In the extended model, its components (age, LN metastasis, ETE, distant metastasis, and tumor size) were taken individually and each variable was analyzed. Schoenfeld Enterprises was used in both models. Sensitivity analysis was conducted by applying a backward elimination approach to a multivariate Cox regression with all univariate explanatory variables. Statistical analyses were performed using SAS version 9.4 (SAS Institute Inc, Cary, NC, USA) and R 3.4.3 (Vienna, Austria; http://www.R-project.org). A *p*-value ≤ 0.05 was considered statistically significant.

## 5. Conclusions

When analyzed with the variables of TNM-8, *TERT* promoter mutations acted as an independent prognostic factor after adjusting for the conventional clinicopathological risk factors and stage, regardless of the histological types or stage, increasing prognostic predictability, and suggesting that this parameter is an indicator of poor prognosis. Therefore, subsequent research is required to propose a new staging system that combines the *TERT* mutational state with conventional clinicopathological risk factors.

## Figures and Tables

**Figure 1 cancers-13-00648-f001:**
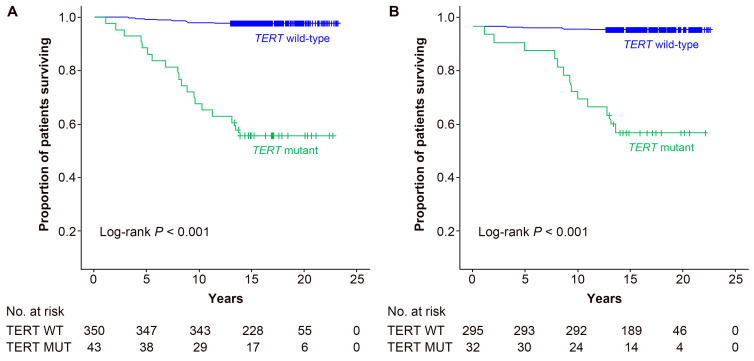
Kaplan–Meier curves of thyroid cancer-specific survival, according to *TERT* promoter mutational status in patients with (**A**) differentiated thyroid cancer and (**B**) papillary thyroid cancer.

**Figure 2 cancers-13-00648-f002:**
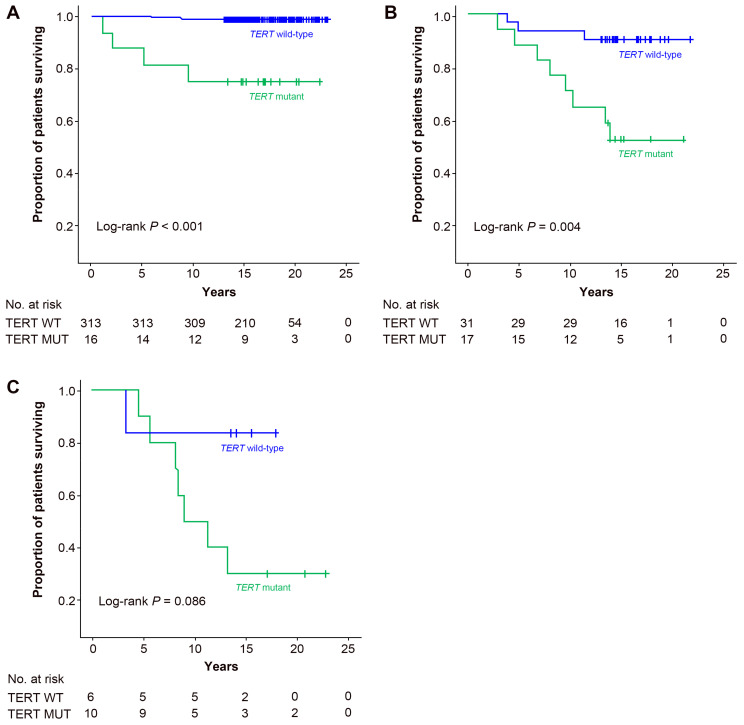
Kaplan–Meier curves of thyroid cancer-specific survival based on *TERT* promoter mutational status, according to the stage at diagnosis in patients with differentiated thyroid cancer (DTC). (**A**) stage I, (**B**) stage II, and (**C**) stages III and IV.

**Table 1 cancers-13-00648-t001:** Clinicopathologic and genetic characteristics of patients with DTC and PTC.

Characteristics	DTC (*n* = 393), N (%)	PTC (*n* = 327), N (%)
Sex	-	-
Female	329 (83.7)	276 (84.4)
Male	64 (16.3)	51 (15.6)
Age, years	-	-
Median	43	43
Range	16–81	16–81
<55	319 (81.2)	265 (81.0)
≥55	74 (18.8)	62 (19.0)
*TERT* promoter status	-	-
Wild-type	350 (89.1)	295 (90.2)
Mutation	43 (10.9)	32 (9.8)
*BRAF* mutation	-	-
Wild-type	117 (37.0)	65 (24.6)
Mutation	199 (63.0)	199 (75.4)
Missing data	77	63
Histologic type	-	-
PTC	327 (83.2)	-
FTC	66 (16.8)	-
Multifocality	-	-
Absent	286 (72.8)	228 (69.7)
Present	107 (27.2)	99 (30.3)
Lymph node metastasis	-	-
Absent	199 (50.6)	135 (41.3)
Present	193 (49.1)	191 (58.4)
Missing data	1 (0.3)	1 (0.3)
Extrathyroidal extension	-	-
Absent	352 (89.6)	290 (88.7)
Present	41 (10.4)	37 (11.3)
Distant metastasis	-	-
Absent	370 (94.1)	313 (95.7)
Present	23 (5.9)	14 (4.3)
Stage at diagnosis ^a^	-	-
I	329 (83.7)	274 (83.8)
II	48 (12.2)	42 (12.8)
III and IV	16 (4.1)	11 (3.4)
Tumor size, cm	-	-
Median	2.7	2.5
Range	0.4–12.0	0.4–10.5
<2.0	45 (11.5)	35 (10.7)
2.0–4.0	293 (74.6)	253 (77.4)
>4.0	55 (14.0)	39 (11.9)
RAI total dose, mCi	-	-
Median	130	160
Range	0–1400	0–1250

Abbreviations: *TERT*, telomerase reverse transcriptase; *BRAF*, v-Raf murine sarcoma viral oncogene homolog B; DTC, differentiated thyroid cancer; PTC, papillary thyroid cancer; FTC, follicular thyroid cancer; RAI, radioactive iodine. ^a^ Staging according to the American Joint Committee on Cancer Thyroid Cancer Staging System, 8th edition, 2016.

**Table 2 cancers-13-00648-t002:** Associations between TERT mutation status and clinicopathological variables in patients with DTC.

Variables	DTC			
	*TERT* WT, N (%)	*TERT* Mut, N (%)	OR ^a^ (95%CI)	*p*-Value
Sex	-	-	-	-
Female	293 (89.1)	36 (10.9)	1.00 (reference)	0.999
Male	57 (89.1)	7 (10.9)	1.00 (0.39–2.23)	-
Age, years	-	-	-	-
Per 5 years	-	-	1.94 (1.64–2.35)	<0.001
<55	304 (95.3)	15 (4.7)	1.00 (reference)	<0.001
≥55	46 (62.2)	28 (37.8)	12.34 (6.22–25.39)	-
*BRAF* mutation	-	-	-	-
Wild-type	107 (91.5)	10 (8.5)	1.00 (reference)	0.333
Mutant	175 (87.9)	24 (12.1)	1.47 (0.69–3.33)	-
Missing data	68 (88.3)	9 (11.7)	-	-
Histologic type	-	-	-	-
PTC	295 (90.2)	32 (9.8)	1.00 (reference)	0.107
FTC	55 (83.3)	11 (16.7)	1.84 (0.84–3.78)	-
Multifocality	-	-	-	-
Absent	255 (89.2)	31 (10.8)	1.00 (reference)	0.915
Present	95 (88.8)	12 (11.2)	1.04 (0.49–2.06)	-
Lymph node metastasis	-	-	-	-
Absent	179 (89.9)	20 (10.1)	1.00 (reference)	0.666
Present	171 (88.6)	22 (11.4)	1.15 (0.61–2.20)	-
Missing data	0 (0.0)	1 (100)	-	-
Extrathyroidal extension	-	-	-	-
Absent	322 (91.5)	30 (8.5)	1.00 (reference)	<0.001
Present	28 (68.3)	13 (31.7)	4.98 (2.29–10.51)	-
Distant metastasis	-	-	-	-
Absent	335 (90.5)	35 (9.5)	1.00 (reference)	0.001
Present	15 (65.2)	8 (34.8)	5.10 (1.94–12.64)	-
Stage at diagnosis ^b^	-	-	-	-
I	313 (95.1)	16 (4.9)	1.00 (reference)	<0.001
II	31 (64.6)	17 (35.4)	10.73 (4.94–23.56)	<0.001
III and IV	6 (37.5)	10 (62.5)	32.60 (10.81–107.06)	<0.001
Tumor size, cm	-	-	-	-
<2.0	41 (91.1)	4 (8.9)	1.00 (reference)	0.375
2.0–4.0	263 (89.8)	30 (10.2)	1.17 (0.43–4.09)	0.779
>4.0	46 (83.6)	9 (16.4)	2.01 (0.60–7.85)	0.275
RAI total dose, mCi	-	-	-	-
Median (range)	130 (0–1200)	230 (0–1400)	1.002 (1.001–1.003)	<0.001

Abbreviations: *TERT*, telomerase reverse transcriptase; *BRAF*, v-Raf murine sarcoma viral oncogene homolog B; DTC, differentiated thyroid cancer; PTC, papillary thyroid cancer; FTC, follicular thyroid cancer; RAI, radioactive iodine. ^a^ The odds ratio (OR) represents the odds for being a *TERT* promoter mutation-carrier compared with being a *TERT* promoter wild-type-carrier. ^b^ Staging according to the American Joint Committee on Cancer Thyroid Cancer Staging System, 8th edition, 2016.

**Table 3 cancers-13-00648-t003:** Associations between *TERT* mutation status or clinicopathological variables and thyroid cancer-specific survival in patients with DTC.

Variables	N ^a^	10-Year Survival Rate (%)	Univariate Cox Models		Multivariate Cox Model 1 ^b^ (Extended Model)		Multivariate Cox Model 2 ^c^ (Restricted Model)	
			HR (95% CI)	*p*-Value	HR (95% CI)	*p*-Value	HR (95% CI)	*p*-Value
Sex	-	-	-	-	-	-	-	-
Female	329	95.1	1.00 (reference)	-	1.00 (reference)	0.099	1.00 (reference)	0.054
Male	64	92.2	1.83 (0.78–4.34)	0.167	2.27 (0.86–5.98)	-	2.46 (0.99–6.15)	-
Age, years ^d^	-	-	-	-	-	-	-	-
Per 5 years	393	-	1.75 (1.50–2.06)	<0.001	1.46 (1.18–1.79)	<0.001	-	-
<55	319	97.5	1.00 (reference)	<0.001	-	-	-	-
≥55	74	82.4	11.34 (4.96–25.93)	-	-	-	-	-
*TERT* mutation	-	-	-	-	-	-	-	-
Wild-type	350	98.0	1.00 (reference)	<0.001	1.00 (reference)	0.002	1.00 (reference)	<0.001
Mutant	43	67.4	24.15 (10.56–55.25)	-	5.18 (1.81–14.82)	-	9.93 (3.67–26.90)	-
*BRAF* mutation	-	-	-	-	-	-	-	-
Wild-type	117	92.3	1.00 (reference)	0.331	-	-	-	-
Mutant	199	97.0	0.64 (0.26–1.57)	-	-	-	-	-
Histologic type	-	-	-	-	-	-	-	-
PTC	327	96.6	1.00 (reference)	0.004	1.00 (reference)	0.028	1.00 (reference)	0.001
FTC	66	84.8	3.15 (1.44–6.88)	-	3.02 (1.13–8.09)	-	4.11 (1.77–9.54)	-
Multifocality	-	-	-	-	-	-	-	-
Absent	286	94.4	1.00 (reference)	0.847	-	-	-	-
Present	107	95.3	0.92 (0.39–2.17)	-	-	-	-	-
Lymph node metastasis	-	-	-	-	-	-	-	-
Absent	199	94.0	1.00 (reference)	0.964	-	-	-	-
Present	193	95.3	1.02 (0.47–2.20)	-	-	-	-	-
Extrathyroidal extension	-	-	-	-	-	-	-	-
Absent	352	96.0	1.00 (reference)	<0.001	1.00 (reference)	0.227	-	-
Present	41	82.9	4.78 (2.14–10.64)	-	1.81 (0.69–4.75)	-	-	-
Distant metastasis	-	-	-	-	-	-	-	-
Absent	370	96.8	1.00 (reference)	<0.001	1.00 (reference)	0.063	-	-
Present	23	60.9	10.85 (4.86–24.22)	-	2.61 (0.95–7.20)	-	-	-
Stage at diagnosis ^e^	-	-	-	-	-	-	-	-
I	329	97.6	1.00 (reference)	<0.001	-	-	1.00 (reference)	0.003
II	48	85.4	10.42 (4.19–25.92)	<0.001	-	-	5.00 (1.73–14.50)	0.003
III and IV	16	62.5	26.82 (10.03–71.70)	<0.001	-	-	6.57 (2.04–21.12)	0.002
Tumor size, cm	-	-	-	-	-	-	-	-
<2.0	45	95.6	1.00 (reference)	0.014	1.00 (reference)	0.298	-	-
2.0–4.0	293	95.9	1.24 (0.28–5.38)	0.776	0.77 (0.16–3.65)	0.744	-	-
>4.0	55	87.3	3.96 (0.86–18.34)	0.078	1.61 (0.30–8.67)	0.582	-	-

Abbreviations: *TERT*, telomerase reverse transcriptase; *BRAF*, v-Raf murine sarcoma viral oncogene homolog B; DTC, differentiated thyroid cancer; PTC, papillary thyroid cancer; FTC, follicular thyroid cancer; HR, hazard ratio. ^a^ The number based on available data for a particular variable in the univariate analysis. ^b^ Model in which all predictors with univariate *p*-values ≤ 0.20 were included; no interactions were considered. ^c^ Restricted model that includes “*TERT*” and “Stage at diagnosis”; no interactions were considered. ^d^ Multivariate Cox regression analysis results for “Age” were analyzed and presented for the continuous linear variable. ^e^ Staging according to the American Joint Committee on Cancer Thyroid Cancer Staging System, 8th edition, 2016. HR is referred to as the risk of thyroid cancer-specific death.

**Table 4 cancers-13-00648-t004:** Associations between *TERT* mutation status or clinicopathologic variables and thyroid cancer-specific survival in patients with PTC.

Variables	N ^a^	10-Year Survival Rate (%)	Univariate Cox Models		Multivariate Cox Model 1 ^b^ (Extended Model)		Multivariate Cox Model 2 ^c^ (Restricted Model)	
			HR (95% CI)	*p*-Value	HR (95% CI)	*p*-Value	HR (95% CI)	*p*-Value
Sex	-	-	-	-	-	-	-	-
Female	276	97.5	1.00 (reference)	0.026	1.00 (reference)	0.244	1.00 (reference)	0.072
Male	51	92.2	3.09 (1.14–8.36)	-	2.07 (0.61–7.02)	-	2.57 (0.92–7.22)	-
Age, years ^d^	-	-	-	-	-	-	-	-
Per 5 years	327	-	1.73 (1.42–2.10)	<0.001	1.30 (0.99–1.70)	0.059	-	-
<55	265	98.1	1.00 (reference)	<0.001	-	-	-	-
≥55	62	90.3	11.00 (3.87–31.23)	-	-	-	-	-
*TERT* mutation	-	-	-	-	-	-	-	-
Wild-type	295	99.0	1.00 (reference)	<0.001	1.00 (reference)	0.002	1.00 (reference)	<0.001
Mutant	32	75.0	36.27 (11.81–111.42)	-	10.68 (2.36–48.27)	-	18.55 (4.83–71.18)	-
*BRAF* mutation	-	-	-	-	-	-	-	-
Wild-type	65	98.5	1.00 (reference)	0.253	-	-	-	-
Mutant	199	97.0	3.31 (0.42–25.89)	-	-	-	-	-
Multifocality	-	-	-	-	-	-	-	-
Absent	228	96.9	1.00 (reference)	0.664	-	-	-	-
Present	99	96.0	1.25 (0.46–3.37)	-	-	-	-	-
Lymph node metastasis	-	-	-	-	-	-	-	-
Absent	135	97.0	1.00 (reference)	0.410	-	-	-	-
Present	191	96.3	1.56 (0.54–4.49)	-	-	-	-	-
Extrathyroidal extension	-	-	-	-	-	-	-	-
Absent	290	97.6	1.00 (reference)	0.002	1.00 (reference)	0.230	-	-
Present	37	89.2	4.67 (1.73–12.65)	-	2.06 (0.63–6.70)	-	-	-
Distant metastasis	-	-	-	-	-	-	-	-
Absent	313	97.4	1.00 (reference)	0.007	1.00 (reference)	0.199	-	-
Present	14	78.6	5.59 (1.61–19.47)	-	2.85 (0.58–14.12)	-	-	-
Stage at diagnosis ^e^	-	-	-	-	-	-	-	-
I	274	98.2	1.00 (reference)	<0.001	-	-	1.00 (reference)	0.221
II	42	92.9	9.60 (3.05–30.25)	<0.001	-	-	2.65 (0.72–9.78)	0.142
III and IV	11	72.7	31.06 (8.97–107.58)	<0.001	-	-	3.45 (0.79–15.07)	0.099
Tumor size, cm	-	-	-	-	-	-	-	-
<2.0	35	97.1	1.00 (reference)	0.087	1.00 (reference)	0.486	-	-
2.0–4.0	253	97.2	1.53 (0.20–11.84)	0.685	0.75 (0.09–6.19)	0.790	-	-
>4.0	39	92.3	4.70 (0.55–40.19)	0.158	1.57 (0.15–16.21)	0.706	-	-

Abbreviations: *TERT*, telomerase reverse transcriptase; *BRAF*, v-Raf murine sarcoma viral oncogene homolog B; PTC, papillary thyroid cancer; FTC, follicular thyroid cancer; HR, hazard ratio ^a^ The number based on available data for a particular variable in the univariate analysis. ^b^ Model in which all predictors with univariate *p*-values ≤ 0.20 were included; no interactions were considered. ^c^ Restricted model that includes “*TERT*” and “Stage at diagnosis”; no interactions were considered. ^d^ Multivariate Cox regression analysis results for “Age” were analyzed and presented for the continuous linear variable. ^e^ Staging according to the American Joint Committee on Cancer Thyroid Cancer Staging System, 8th edition, 2016. HR is referred to as the risk of thyroid cancer-specific death.

## Data Availability

The datasets generated during and/or analyzed during the current study are available from the corresponding author on reasonable request.

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
