# Peer review of "TERT Promoter Mutations and the 8th Edition TNM Classification in Predicting the Survival of Thyroid Cancer Patients"

_cancers, 2021, doi:10.3390/cancers13040648_

Round 1

Reviewer 1 Report

The authors have significantly improved the previous vesrion of the manuscript, applying to the reviewers' comments. I consider the topic and results of the paper to be clinically important.

Author Response

Thank you very much for the helpful comments and suggestions, again.

Reviewer 2 Report

The paper is acceptable in this form

Author Response

(The authors gave the same response as above.)

Reviewer 3 Report

The authors have approached issues raised to satisfaction.

Author Response

(The authors gave the same response as above.)

Reviewer 4 Report

The Authors agreed to address the issue concerning the downstaging of patients, but this reviewer has major concerns about the appropriateness of revision. Specifically:

1) the Authors performed the required new analysis but the new data were only mentioned in the Discussion and not reported in the Results Section. Moreover, the p-values of each comparison were not provided

2) The subdivision of patients in 4 groups does not seem to be appropriate, since some patients of groups 1 and 2 were also included in groups 3 or 4. This Reviewer suggests to keep separate the analyses of downstaged  vs not dowstaged from the analysis of downstaged for age vs downstaged for tumor features. Performing separately these analyses, it may be that mutant TERT was significantly more frequent in downstaged vs. not downstaged, whereas likely it was not significantly associated with the type of downstaging (please, could you provide the p-values of comparisons).

Author Response

Reviewers' comments:

Reviewer: 4

Comments to the Author
The Authors agreed to address the issue concerning the downstaging of patients, but this reviewer has major concerns about the appropriateness of revision. Specifically:

  • The Authors performed the required new analysis but the new data were only mentioned in the Discussion and not reported in the Results Section. Moreover, the p-values of each comparison were not provided

Answer] The problem of TERT mutation and downstaging presented by the reviewer is considered a clinically important point. The role of TERT mutations in identifying the subjects in the TNM 8th stage II who were at higher risk of cancer-related death can be inferred from the current results (Figure 2B). The authors believe that the added analysis would strengthen the clinical implication of the study results. Following sentences were added in Results section.

[Result section, page 5, 1st paragraph]:

In addition, in the subgroup analysis conducted based on each stage, the survival rate of patients with TERT mutations was significantly lower than that for individuals with wild-type TERT (P < 0.001 in stage I, P = 0.004 in stage II, P = 0.086 in stage III & IV; Figure 2). Since subjects in stage II showed a significantly lower survival as compared to stage I (Table 3), we stratified the stage II subjects in those who were downstaged vs those not downstaged by TNM 8th edition (Table S1). The mutant TERT had a tendency to be more frequent in downstaged vs. not downstaged (p = 0.074, statistically not significant), suggesting that in several of these cases the downstaging may have led toward an underestimation of cancer aggressivity.

Supplemental Table S1. Subgroup analysis of stage II patients according to downstaging by TNM-8.

N, (%)

Group 1a

(N = 10)

Group 2 b

(N = 38)

P-value

MUT TERT

1 (10)

16 (42.1)

0.074

Cancer-specific death

1 (10)

10 (26.3)

0.416

Cancer-specific death with MUT TERT

0

8 (80.0)

0.273

a Group 1 : TNM-8 Stage II patients who were also Stage II in TNM-7

b Group 2 : TNM-8 Stage II Patients who were downstaged by TNM-8

Abbreviations: TNM-8, the American Joint Committee on Cancer Thyroid Cancer Staging System 8th edition; MUT, mutant; TERT, telomerase reverse transcriptase.

  • The subdivision of patients in 4 groups does not seem to be appropriate, since some patients of groups 1 and 2 were also included in groups 3 or 4. This Reviewer suggests to keep separate the analyses of downstaged vs not dowstaged from the analysis of downstaged for age vs downstaged for tumor features. Performing separately these analyses, it may be that mutant TERT was significantly more frequent in downstaged vs. not downstaged, whereas likely it was not significantly associated with the type of downstaging (please, could you provide the p-values of comparisons).

Answer] The authors do appreciate the reviewer’s valuable comments. I fully agree with the reviewer’s opinion. We analyzed downstaged vs. not downstaged and downstaged for age vs. downstaged for tumor features separately and expressed them in each supplemental table. At this time, the analysis of all downstaged patients was judged to obscure the point of the entire paper and we focused only on Stage II. I ask for the reviewer’s understanding about this politely.

[Result section, page 5, 1st paragraph]:

In addition, in the subgroup analysis conducted based on each stage, the survival rate of patients with TERT mutations was significantly lower than that for individuals with wild-type TERT (P < 0.001 in stage I, P = 0.004 in stage II, P = 0.086 in stage III & IV; Figure 2). Since subjects in stage II showed a significantly lower survival as compared to stage I (Table 3), we stratified the stage II subjects in those who were downstaged vs those not downstaged by TNM 8th edition (Table S1). The mutant TERT had a tendency to more frequent in downstaged vs. not downstaged (p = 0.074, statistically not significant), suggesting that in several of these cases the downstaging may have led toward an underestimation of cancer aggressivity. Additionally, the frequency of mutant TERT was not associated with the cause of downstaging (age vs tumor classification) among downstaged stage II patients (p = 0.682) (Table S2).

Supplemental Table S2. Subgroup analysis of downstaged stage II patients according to the cause of downstaging in TNM-8.

N, (%)

Group 2Aa

(N = 6)

Group 2B b

(N = 32)

P-value

MUT TERT

3 (50)

13 (40.6)

0.682

Cancer-specific death

4 (66.7)

6 (18.8)

0.031

Cancer-specific death with MUT TERT

3 (75)

5 (83.3)

1.000

a Group 2A : TNM-8 Stage II Patients who were downstaged by TNM-8 according to age

b Group 2B: TNM-8 Stage II Patients who were downstaged by TNM-8 according to tumor classification

Abbreviations: TNM-8, the American Joint Committee on Cancer Thyroid Cancer Staging System 8th edition; MUT, mutant; TERT, telomerase reverse transcriptase.

Thank you very much for the helpful comments and suggestions. We did our best to address the reviewer’s requests and hope that it would be the satisfied version for the publication.

This manuscript is a resubmission of an earlier submission. The following is a list of the peer review reports and author responses from that submission.

Round 1

Reviewer 1 Report

This paper is an update of a previous study, looking at TERT promoter mutation in the context of DTC/PTC, in connection to the AJCC TNM 8th edition. Again, it is shown that TERT mutation analysis adds prognostic information to other variables.

The paper is well written and is not without interest to readers in the thyroid cancer field. However, it is almost a copy-paste version of a previous paper from the same group published in ERC 2016. This (that this is just an update of a previous study) should be pointed out more clearly, preferably already in the Abstract, or why not already in the title?

It is stated in the Summary that "Since the current AJCC TNM-8 is insufficient to distinguish the risk of mortality in patients with DTC...." etc. However, it is also stated in the Introduction that "After the staging system changed to TNM-8, the predictability of cancer-specific survival has improved for DTC patients..." etc. This is clearly contradictory. Which position Re this question is the authors'?

In addition, in the Discussion, the authors could mention novel methods to identify TERT mutations, such as digital droplet PCR, its pros and cons, in the light of the older detection method used herein.

Author Response

Reviewers' comments:

Reviewer: 1

Comments to the Author
This paper is an update of a previous study, looking at TERT promoter mutation in the context of DTC/PTC, in connection to the AJCC TNM 8th edition. Again, it is shown that TERT mutation analysis adds prognostic information to other variables.

1. The paper is well written and is not without interest to readers in the thyroid cancer field. However, it is almost a copy-paste version of a previous paper from the same group published in ERC 2016. This (that this is just an update of a previous study) should be pointed out more clearly, preferably already in the Abstract, or why not already in the title?

Answer] The authors do appreciate the reviewer’s generous and helpful comments.

I explained only to the method section that this research is an updated version of the previous study (Kim et al. ERC 2016) because I was worried that the title or abstract would be too long, but I added the descriptions to the abstract in response to the reviewer’s comment. In addition, some sentences were rewritten to clarify the enrolled patients and the discrepancies between the manuscript and the article previously published.

[Abstract section, page 1]:

Our research group has previously shown that the presence of TERT promoter mutations is an independent prognostic factor by applying the TERT mutation status to the variables of AJCC 7th edition. This study aimed to determine if TERT mutations could be independent predictors of thyroid cancer-specific mortality based on the AJCC TNM 8th edition with long-term follow-up.

[Discussion section, page 9, 2nd paragraph]:

In our previous report, we analyzed the survival rate including TERT mutation status based on TNM-7 [15]. In this study, we performed survival analysis by restaging the 393 DTC patients in the cohort from the previous study using TNM-8 guidelines and by extending the follow-up period to determine whether TERT mutations are still a factor associated with poor prognosis.

[Materials and Methods section, page 9, 6rd paragraph]:

A total of 393 patients who were pathologically diagnosed with DTC, including 327 PTC and 66 FTC cases (including Hurthle cell carcinoma), after thyroidectomy and neck dissection between October 1994 and December 2004 were included in this study, all of which were enrolled in previous studies.

2. It is stated in the Summary that "Since the current AJCC TNM-8 is insufficient to distinguish the risk of mortality in patients with DTC...." etc. However, it is also stated in the Introduction that "After the staging system changed to TNM-8, the predictability of cancer-specific survival has improved for DTC patients..." etc. This is clearly contradictory. Which position Re this question is the authors'?

Answer] We agree with the reviewer’s opinion. The author's intention was to emphasize that the AJCC TNM 8th edition was not the best risk certification system enough, although the mortality predicting power increased as it was updated, but it seems that the explanation was insufficient. Following sentences and references were added in introduction section to address this issue.

[Introduction section, page 2, 1st paragraph]:

The major changes in TNM-8 include the advanced age cutoff from 45 to 55 years [6]. In addition, minimal extrathyroidal extension (ETE) has been excluded from the T3 definition. Level VII lymph nodes (LNs) were reclassified as central neck LNs, N1 disease was not staged up to stage III, and distant metastases in older patients were changed to stage IVB for DTC [7]. As a result, nearly 30% of DTC patients were downstaged by TNM-8 [8]. After the staging system changed to TNM-8, the predictability of cancer-specific survival (CSS) has improved for DTC and PTC patients, but it has not improved for FTC patients [9]. In addition, Manzardo et al. recently reported that the risk of structural recurrence of DTC patients downstaged by tumor classification in TNM-8 may be overlooked [8], and the concept of molecular profile that has recently emerged in the prognosis is still missing [10].

[Reference section]

  1. Manzardo, O.A.; Cellini, M.; Indirli, R.; Dolci, A.; Colombo, P.; Carrone, F.; Lavezzi, E.; Mantovani, G.; Mazziotti, G.; Arosio, M.; et al. TNM 8th edition in thyroid cancer staging: is there an improvement in predicting recurrence? Endocrine-related cancer 2020, 27, 325-336.

  1. Xing, M.; Haugen, B.R.; Schlumberger, M. Progress in molecular-based management of differentiated thyroid cancer. Lancet (London, England) 2013, 381, 1058-1069.

3. In addition, in the Discussion, the authors could mention novel methods to identify TERT mutations, such as digital droplet PCR, its pros and cons, in the light of the older detection method used herein.

Answer] The authors do appreciate the reviewer’s thoughtful comments. However, this study did not specifically consider the various detection methods of TERT promoter mutations because we wanted to emphasize the clinical usefulness of TERT promoter mutations in differentiated thyroid cancer.

Thank you very much for the helpful comments and suggestions. We did our best to address the reviewer’s requests and hope that it would be the satisfied version for the publication.

Reviewer 2 Report

In this retrospective study, Park et al evaluated the role of TERT promoter mutations as biomarker for prediction of mortality in subjects with differentiated thyroid carcinoma. The topic is timely and the results are potentially interesting. The following issues need to be addressed.

  1. The Authors should specify how many subjects enrolled in this study had been already evaluated in their former paper on the topic (ref#13]. This is important, in order to clarify whether this new study was just a post-hoc analysis of the previous one.
  2. The shift of TNM from the 7th to 8th edition leads to downstage a remarkable number of patients (i.e., re-classification of subjects from stages III-IV to stage II). Interestingly, subjects in stage II showed a significantly lower survival as compared to stage I (Table 3), suggesting that in several of these cases the downstaging may have led toward an underestimation of cancer aggressivity. Consistently, in a recent paper (Manzardo et al. Endocrine Related Cancer 2020), the downstaging by TNM 8th edition induced to overlook those individuals predisposed to structural recurrence, potentially causing uncertainty in the therapeutic decision-making at the time of disease’s diagnosis. Based on these considerations, this reviewer would suggest to evaluate the role of TERT mutations in identifying the subjects in the TNM 8th stage II who were at higher risk of cancer-related death. For this purpose, the Authors should stratify the stage II subjects in those who were downstaged vs those not downstaged by TNM 8th edition. Moreover, it may be interesting to evaluate whether the higher rate of TERT mutations might have associated with the type of downstaging (tumor classification vs. age).
  3. Did the authors have any information on the relationship between TERT and risk of recurrence and response to therapy, according to the ATA 2015 criteria?
  4. Table 1: It is unclear why the Authors did the comparison between PTC and DTC. Likely, PTC should have been compared with FTC. Moreover, the p-values of each comparison should be included in the Table.
  5. Tables 3 and 4: the HR are referred to the risk of death; it should be specified
  6. Figure 1: why did the authors compare DTC and PTC rather than FTC and PTC?

Author Response

Reviewers' comments:

Reviewer: 2

Comments to the Author
In this retrospective study, Park et al evaluated the role of TERT promoter mutations as biomarker for prediction of mortality in subjects with differentiated thyroid carcinoma. The topic is timely and the results are potentially interesting. The following issues need to be addressed.

1. The Authors should specify how many subjects enrolled in this study had been already evaluated in their former paper on the topic (ref#13]. This is important, in order to clarify whether this new study was just a post-hoc analysis of the previous one.

Answer] Thank you very much for the thoughtful comment. The DTC patients in this study are totally same as 393 DTC patients who were enrolled in the previous study (Kim et al. ERC 2016). Previous research included PDTC/ATC patients, but this study excluded PDTC/ATC patients, increased the follow up period and added RAI variables, focusing only on DTC patients. Following sentences were rewritten to clarify the enrolled patients and the discrepancies between the manuscript and the article previously published.

[Discussion section, page 9, 2nd paragraph]:

In our previous report, we analyzed the survival rate including TERT mutation status based on TNM-7 [15]. In this study, we performed survival analysis by restaging the 393 DTC patients in the cohort from the previous study using TNM-8 guidelines and by extending the follow-up period to determine whether TERT mutations are still a factor associated with poor prognosis.

[Materials and Methods section, page 9, 6rd paragraph]:

A total of 393 patients who were pathologically diagnosed with DTC, including 327 PTC and 66 FTC cases (including Hurthle cell carcinoma), after thyroidectomy and neck dissection between October 1994 and December 2004 were included in this study, all of which were enrolled in previous studies.

2. The shift of TNM from the 7th to 8th edition leads to downstage a remarkable number of patients (i.e., re-classification of subjects from stages III-IV to stage II). Interestingly, subjects in stage II showed a significantly lower survival as compared to stage I (Table 3), suggesting that in several of these cases the downstaging may have led toward an underestimation of cancer aggressivity. Consistently, in a recent paper (Manzardo et al. Endocrine Related Cancer 2020), the downstaging by TNM 8th edition induced to overlook those individuals predisposed to structural recurrence, potentially causing uncertainty in the therapeutic decision-making at the time of disease’s diagnosis. Based on these considerations, this reviewer would suggest to evaluate the role of TERT mutations in identifying the subjects in the TNM 8th stage II who were at higher risk of cancer-related death. For this purpose, the Authors should stratify the stage II subjects in those who were downstaged vs those not downstaged by TNM 8th edition. Moreover, it may be interesting to evaluate whether the higher rate of TERT mutations might have associated with the type of downstaging (tumor classification vs. age).

Answer] The authors do appreciate the reviewer’s valuable comments. Manzardo's paper has been reviewed and the reviewer’s suggestion is a very interesting idea. Four groups were formed for further analysis in response to the reviewer's opinion.

Of the 48 patients with Stage II in TNM-8,

Group1 (n=10): TNM-8 Stage II patients who were also Stage II in TNM-7

Group2 (n=38): TNM-8 Stage II Patients who were downstaged by TNM-8

Of the 160 downstaged patients in TNM-8,

Group3 (n=83): Patients who were downstaged according to age

Group4 (n=77): Patients who were downstaged according to tumor classification

MUT TERT, N(%)

Cancer-specific death, N(%)

Death with MUT TERT, N(%)

Group 1 (n=10)

1 (10)

1 (10)

0

Group 2 (n=38)

16 (42.1)

10 (26.3)

8 (80.0)

Group 3 (n=83)

11 (13.3)

6 (7.2)

5 (83.3)

Group 4 (n=77)

23 (29.9)

10 (13.0)

10 (100)

Based on the results above, downstaged patients with the same stage II in TNM-8 (Group 2) have a high ratio of TERT mutations and poor prognosis than not downstaged patients (Group 1). In TNM-8, patients with high (III, IV) stage and mutant TERT went down to stage II, and the prognostic difference increased even within the same stage II. This means that the prognostic discrimination within stage II has decreased. In addition, compared to Group 3, the patients in Group 4 have a high rate of mutant TERT and the prognosis is bad. In particular, patients who were downstaged due to changes in tumor classification have a cause of poor prognosis, but it can be overlooked. In these results, it can be seen that the higher rate of TERT mutations has associated with the type of downstaging and poor prognosis.

In this study, we focused on confirming whether TERT promoter mutations still acts as the independent prognostics factor by applying TERT mutation status to the eighth edition of AJCC TNM in DTC patients, so we did not add the above to the manuscript. However, Manzardo’s paper (ERC 2020) was cited in this manuscript to help reinforce existing known knowledge.

Since we have been considering follow-up studies, I think it would be better to increase the number of samples and analyze your proposal in deeply and proceed with the subject of follow-up research of this study.

[Introduction section, page 2, 1st paragraph]:

The major changes in TNM-8 include the advanced age cutoff from 45 to 55 years [6]. In addition, minimal extrathyroidal extension (ETE) has been excluded from the T3 definition. Level VII lymph nodes (LNs) were reclassified as central neck LNs, N1 disease was not staged up to stage III, and distant metastases in older patients were changed to stage IVB for DTC [7]. Therefore, nearly 30% of DTC patients were downstaged by TNM-8 [8]. After the staging system changed to TNM-8, the predictability of cancer-specific survival (CSS) has improved for DTC and PTC patients, but it has not improved for FTC patients [9]. In addition, Manzardo et al. recently reported that the risk of structural recurrence of DTC patients downstaged by TNM-8 may be overlooked [8], and the concept of molecular profile that has recently emerged in the prognosis is still missing [10].

[Reference section]

  1. Manzardo, O.A.; Cellini, M.; Indirli, R.; Dolci, A.; Colombo, P.; Carrone, F.; Lavezzi, E.; Mantovani, G.; Mazziotti, G.; Arosio, M.; et al. TNM 8th edition in thyroid cancer staging: is there an improvement in predicting recurrence? Endocrine-related cancer 2020, 27, 325-336.

3. Did the authors have any information on the relationship between TERT and risk of recurrence and response to therapy, according to the ATA 2015 criteria?

Answer] Thank you for your comment. We have already reported the relationship between TERT and dynamic risk stratification in previous research (Kim et al. ERC 2016, ref#15). The study focused on the relationship between TERT and cancer specific death and did not see the relationship with recurrence or response to therapy.

4. Table 1: It is unclear why the Authors did the comparison between PTC and DTC. Likely, PTC should have been compared with FTC. Moreover, the p-values of each comparison should be included in the Table.

Answer] Since the data analyzed by DTC and PTC groups are displayed in the text, the baseline characteristics of DTC and PTC groups are shown in Table 1. The P-value is not included because it is not intended to compare DTC and PTC groups. In addition, the FTC group has a small sample size and has not been analyzed separately. This is referred to as limitation in the discussion section.

5. Tables 3 and 4: the HR are referred to the risk of death; it should be specified

Answer] Thank you very much for the thoughtful comment. We added the sentences as recommend in footnote of Table 3 and 4.

[Table 3 and 4, footnote]:

HR are referred to the risk of thyroid cancer-specific death.

6. Figure 1: why did the authors compare DTC and PTC rather than FTC and PTC?

Answer] We wanted to analyze the FTC group separately, but the sample size was small, so we did not analyze it separately. Since the AJCC staging system is divided into DTC, MTC, and ATC according to the histologic type, the DTC group was mainly analyzed, and the PTC group, which accounts for more than 90% of the DTC, was further analyzed.

Thank you very much for the helpful comments and suggestions. We did our best to address the reviewer’s requests and hope that it would be the satisfied version for the publication.

Reviewer 3 Report

In this paper the authors describe the effect of TERT mutations in predicting the survival of patients with differentiated thyroid cancers.

The aim of the work has a high impact on the clinical management of these patients. In fact, although most cases have a good prognosis, some patients experience relapse and some of these dies. It’s very important to have the ability to distinguish patients with a poor prognosis from the other. As previously reported by many studies and confirmed by this work, TERT mutation is an independent indicator of poor prognosis.

Minor revision needed:

In Tab. 1 and Tab. 2, the authors should indicate the percentage of BRAF mutation respect to the effective number of analysed samples and not to the total number of cases.

The data of RAI total dose in Tab. 2 is missed.

Author Response

Reviewers' comments:

Reviewer: 3

Comments to the Author
In this paper the authors describe the effect of TERT mutations in predicting the survival of patients with differentiated thyroid cancers.

The aim of the work has a high impact on the clinical management of these patients. In fact, although most cases have a good prognosis, some patients experience relapse and some of these dies. It’s very important to have the ability to distinguish patients with a poor prognosis from the other. As previously reported by many studies and confirmed by this work, TERT mutation is an independent indicator of poor prognosis.

1. In Tab. 1 and Tab. 2, the authors should indicate the percentage of BRAF mutation respect to the effective number of analysed samples and not to the total number of cases.

Answer] We appreciate the helpful comment. We corrected the percentage of BRAF mutation in Table 1. However, Table 2 shows the number of DTC patients with WT BRAF whose TERT is a wild type or mutant (percentage of TERT WT or mutations) and the number of patients with a mutant BRAF whose TERT is a wild type or mutant (percentage of TERT WT or mutations). Therefore, it was not modified because it was not related to BRAF missing data.

Revised Table 1. Clinicopathologic and genetic characteristics of patients with DTC and PTC.

Characteristics

DTC (n = 393), N (%)

PTC (n = 327), N (%)

Sex

Female

329 (83.7)

276 (84.4)

Male

64 (16.3)

51 (15.6)

Age, years

Median

43

43

Range

16–81

16–81

< 55

319 (81.2)

265 (81.0)

≥ 55

74 (18.8)

62 (19.0)

TERT promoter status

Wild-type

350 (89.1)

295 (90.2)

Mutation

43 (10.9)

32 (9.8)

BRAF mutation

Wild-type

117 (37.0)

65 (24.6)

Mutation

199 (63.0)

199 (75.4)

Missing data

77

63

Histologic type

PTC

327 (83.2)

FTC

66 (16.8)

Multifocality

Absent

286 (72.8)

228 (69.7)

Present

107 (27.2)

99 (30.3)

Lymph node metastasis

Absent

199 (50.6)

135 (41.3)

Present

193 (49.1)

191 (58.4)

Missing data

1 (0.3)

1 (0.3)

Extrathyroidal extension

Absent

352 (89.6)

290 (88.7)

Present

41 (10.4)

37 (11.3)

Distant metastasis

Absent

370 (94.1)

313 (95.7)

Present

23 (5.9)

14 (4.3)

Stage at diagnosisa

I

329 (83.7)

274 (83.8)

II

48 (12.2)

42 (12.8)

III & IV

16 (4.1)

11 (3.4)

Tumor size, cm

Median

2.7

2.5

Range

0.4–12.0

0.4–10.5

< 2.0

45 (11.5)

35 (10.7)

2.0–4.0

293 (74.6)

253 (77.4)

> 4.0

55 (14.0)

39 (11.9)

RAI total dose, mCi

Median

130

160

Range

0–1400

0–1250

2. The data of RAI total dose in Tab. 2 is missed.

Answer] We apologize for the mistake. We filled out total RAI doses.

Revised Table 2. Associations between TERT mutation status and clinicopathological variables in patients with DTC.

Variables

DTC

TERT WT, N (%)

TERT mut, N (%)

ORa (95%CI)

P-value

Sex

Female

293 (89.1)

36 (10.9)

1.00 (reference)

0.999

Male

57 (89.1)

7 (10.9)

1.00 (0.39–2.23)

Age, years

Per 5 years

1.94 (1.64–2.35)

< 0.001

< 55

304 (95.3)

15 (4.7)

1.00 (reference)

< 0.001

≥ 55

46 (62.2)

28 (37.8)

12.34 (6.22–25.39)

BRAF mutation

Wild-type

107 (91.5)

10 (8.5)

1.00 (reference)

0.333

Mutant

175 (87.9)

24 (12.1)

1.47 (0.69–3.33)

Missing data

68 (88.3)

9 (11.7)

Histologic type

PTC

295 (90.2)

32 (9.8)

1.00 (reference)

0.107

FTC

55 (83.3)

11 (16.7)

1.84 (0.84–3.78)

Multifocality

Absent

255 (89.2)

31 (10.8)

1.00 (reference)

0.915

Present

95 (88.8)

12 (11.2)

1.04 (0.49–2.06)

Lymph node metastasis

Absent

179 (89.9)

20 (10.1)

1.00 (reference)

0.666

Present

171 (88.6)

22 (11.4)

1.15 (0.61–2.20)

Missing data

0 (0.0)

1 (100)

Extrathyroidal extension

Absent

322 (91.5)

30 (8.5)

1.00 (reference)

< 0.001

Present

28 (68.3)

13 (31.7)

4.98 (2.29–10.51)

Distant metastasis

Absent

335 (90.5)

35 (9.5)

1.00 (reference)

0.001

Present

15 (65.2)

8 (34.8)

5.10 (1.94–12.64)

Stage at diagnosisb

I

313 (95.1)

16 (4.9)

1.00 (reference)

< 0.001

II

31 (64.6)

17 (35.4)

10.73 (4.94–23.56)

< 0.001

III & IV

6 (37.5)

10 (62.5)

32.60 (10.81–107.06)

< 0.001

Tumor size, cm

< 2.0

41 (91.1)

4 (8.9)

1.00 (reference)

0.375

2.0–4.0

263 (89.8)

30 (10.2)

1.17 (0.43–4.09)

0.779

> 4.0

46 (83.6)

9 (16.4)

2.01 (0.60–7.85)

0.275

RAI total dose, mCi

Median (range)

130 (0-1200)

230 (0-1400)

1.002 (1.001–1.003)

< 0.001

Thank you very much for the helpful comments and suggestions. We did our best to address the reviewer’s requests and hope that it would be the satisfied version for the publication.

Reviewer 4 Report

The authors analyse the prognostic significance of the presence of pTERT mutations in the tumour for predicting poor prognosis in thyroid cancer in the context of new AJCC TNM 8th Edition.  However, it is not clear, what is the relation of this analysis to the analysis presented by them in their earlier article by Kim et al. Endocrine-Related Cancer 2016, 23, 813-823.

Although the authors cite their article, they do not state, whether they have partially included the same patients, who have already been analysed.

Therefore, it is necessary to clarify the discrepancies between the manuscript and the article previously published.

Author Response

Reviewers' comments:

Reviewer: 4

Comments to the Author
The authors analyse the prognostic significance of the presence of pTERT mutations in the tumour for predicting poor prognosis in thyroid cancer in the context of new AJCC TNM 8th Edition.

1. However, it is not clear, what is the relation of this analysis to the analysis presented by them in their earlier article by Kim et al. Endocrine-Related Cancer 2016, 23, 813-823. Although the authors cite their article, they do not state, whether they have partially included the same patients, who have already been analysed. Therefore, it is necessary to clarify the discrepancies between the manuscript and the article previously published.

Answer] The authors do appreciate the reviewer’s generous and helpful comments. I explained only to the method section that this research is an updated version of the previous study (Kim et al. ERC 2016) because I was worried that the title or abstract would be too long, but I added the descriptions to the abstract in response to the reviewer’s comment.

[Abstract section, page 1]:

Our research group has previously shown that the presence of TERT promoter mutations is an independent prognostic factor by applying the TERT mutation status to the variables of AJCC 7th edition. This study aimed to determine if TERT mutations could be independent predictors of thyroid cancer-specific mortality based on the AJCC TNM 8th edition with long-term follow-up.

In addition, the DTC patients in this study are totally same as 393 DTC patients who were enrolled in the previous study (Kim et al. ERC 2016). Previous research included PDTC/ATC patients, but this study excluded PDTC/ATC patients, increased the follow up period and added RAI variables, focusing only on DTC patients. Following sentences were rewritten to clarify the enrolled patients and the discrepancies between the manuscript and the article previously published.

[Discussion section, page 9, 2nd paragraph]:

In our previous report, we analyzed the survival rate including TERT mutation status based on TNM-7 [15]. In this study, we performed survival analysis by restaging the 393 DTC patients in the cohort from the previous study using TNM-8 guidelines and by extending the follow-up period to determine whether TERT mutations are still a factor associated with poor prognosis.

[Materials and Methods section, page 9, 6rd paragraph]:

A total of 393 patients who were pathologically diagnosed with DTC, including 327 PTC and 66 FTC cases (including Hurthle cell carcinoma), after thyroidectomy and neck dissection between October 1994 and December 2004 were included in this study, all of which were enrolled in previous studies.

Thank you very much for the helpful comments and suggestions. We did our best to address the reviewer’s requests and hope that it would be the satisfied version for the publication.

Round 2

Reviewer 2 Report

The issue concerning the analysis of TERT mutations in relationship with downstaging of patients was not properly addressed by the Authors. This Reviewer considers necessary the inclusion of the new analyses in the revised manuscript.

Reviewer 4 Report

The Authors responded to my questions sufficiently and they enhanced their manuscript considerably.